# Repetitive Transcranial Magnetic Stimulation of the Human Motor Cortex Modulates Processing of Heat Pain Sensation as Assessed by the Offset Analgesia Paradigm

**DOI:** 10.3390/jcm12227066

**Published:** 2023-11-13

**Authors:** Giuseppe Cosentino, Elisa Antoniazzi, Camilla Cavigioli, Vanessa Tang, Giulia Tammam, Chiara Zaffina, Cristina Tassorelli, Massimiliano Todisco

**Affiliations:** 1Department of Brain and Behavioral Sciences, University of Pavia, 27100 Pavia, Italy; 2Translational Neurophysiology Research Unit, IRCCS Mondino Foundation, 27100 Pavia, Italy; 3Headache Science and Neurorehabilitation Center, IRCCS Mondino Foundation, 27100 Pavia, Italy

**Keywords:** heat, modulation, offset analgesia, pain, primary motor cortex, repetitive Transcranial Magnetic Stimulation

## Abstract

Offset analgesia (OA), which is defined as a disproportionately large reduction in pain perception following a small decrease in a heat stimulus, quantifies temporal aspects of endogenous pain modulation. In this study on healthy subjects, we aimed to (i) determine the Heat Pain Threshold (HPT) and the response to constant and dynamic heat stimuli assessing sensitization, adaptation and OA phenomena at the thenar eminence; (ii) evaluate the effects of high-frequency repetitive Transcranial Magnetic Stimulation (rTMS) of the primary motor cortex (M1) on these measures. Twenty-four healthy subjects underwent quantitative sensory testing before and after active or sham 10 Hz rTMS (1200 stimuli) of the left M1, during separate sessions. We did not observe any rTMS-related changes in the HPT or visual analogue scale (VAS) values recorded during the constant trial. Of note, at baseline, we did not find OA at the thenar eminence. Only after active rTMS did we detect significantly reduced VAS values during dynamic heat stimuli, indicating a delayed and attenuated OA phenomenon. rTMS of the left M1 may activate remote brain areas that belong to the descending pain modulatory and reward systems involved in the OA phenomenon. Our findings provide insights into the mechanisms by which rTMS of M1 could exert its analgesic effects.

## 1. Introduction

Repetitive Transcranial Magnetic Stimulation (rTMS) has recently emerged as a safe, non-invasive tool for treating various pain conditions [1,2,3]. Evidence has been provided that both single and multiple sessions of high-frequency rTMS applied to the primary motor cortex (M1) can lead to a reduction in pain perception in patients with chronic pain [4,5,6,7]. Nonetheless, the mechanisms of pain relief remain poorly understood. An increased motor corticospinal excitability induced by high-frequency rTMS has been associated with a more efficient inhibitory pain modulation in healthy subjects [8]. It has been supposed that rTMS could activate the thalamus, suppressing the transmission of sensory information via the spinothalamic pathway [9,10]. According to this hypothesis, a significant increase in the Heat Pain Threshold (HPT) following a single session of rTMS over M1 has been observed in patients with chronic pain [4]. Another potential mechanism through which rTMS over the motor cortex is thought to exert its antinociceptive effects is the activation of specific brain areas that belong to the descending pain modulatory (brainstem, anterior cingulate and dorsolateral prefrontal cortices) and reward systems (putamen, medial prefrontal cortex, and nucleus accumbens) [3,11,12]. It is noteworthy, in this regard, that a connection from M1 to the nucleus accumbens reward circuitry, capable to suppress negative emotional and behavioural aspects associated with neuropathic pain, has been recently unravelled [13].

In this study, we investigated the effect of a single session of high-frequency rTMS applied to M1 on the HPT and offset analgesia (OA) paradigm in a group of healthy subjects. OA is a psychophysical phenomenon defined as a disproportionately large decrease in pain sensation following a tiny decrease in a heat pain stimulus applied over the skin [14,15]. Though the physiological mechanisms underlying OA are not completely understood, activation of both descending pain modulatory and reward systems is supposed to be involved [14,16].

The choice to stimulate the hand area of the left M1 was motivated by several factors: (i) this region represents the main target for treating bilateral chronic pain by rTMS [1,5,17]; (ii) motor threshold values can be easily assessed in the hand muscles, so that the stimulation intensity can be individually dosed [1,17]; (iii) a reliable coil placement can be achieved also without magnetic resonance imaging (MRI)-guided neuronavigation [17]. The OA paradigm was assessed at the thenar eminence of the contralateral right hand, considering that optimal analgesic effects can be achieved when rTMS is applied over the corresponding somatotopic representation (within M1) of the painful territory [18].

In most studies investigating OA, thermal stimuli have been applied to the skin of the forearm or leg. It is unclear whether OA can be similarly observed in other body areas, and one single study failed to detect OA in the hand thenar [19]. In this latter study, however, a shorter than usual time window (i.e., 10 s vs. 20 s) was used to detect OA, and a constant trial was not used to assess the effects of sensitization or adaptation phenomena. The greater thickness of the epidermis in the glabrous skin of the hand could be responsible for a delay and attenuation of the temperature waveform at nociceptor level when conductive heating is applied [20,21,22]. Moreover, differences in nociceptor innervation between the thenar eminence and non-glabrous skin areas could affect responses to both constant and dynamic thermal stimuli [19,20,23].

Based on the above considerations, we aimed to (i) assess OA in the right thenar eminence by using a standard 30 s OA and constant trial paradigm; (ii) investigate whether high-frequency rTMS applied over the hand area of the left M1 could increase the HPT; and (iii) evaluate whether high-frequency rTMS could allow the detection of OA (if it is not detectable or masked by sensitization phenomena at baseline) or potentiate it (if OA is already detected at baseline) in the right hand of healthy subjects.

## 2. Material and Methods

### 2.1. Subjects

We investigated 24 right-handed healthy volunteers (13 females; mean age ± standard deviation: 26 ± 6.9 years; age range: 20–55 years), all naïve for rTMS. All subjects had no history of chronic pain (including recurrent headaches or injury-related pain) or any neurological disorder as assessed by a thorough medical history evaluation. Women were not examined during their menstrual cycle, and we paid attention to carrying out the 3 experimental sessions in the same period of the menstrual cycle in the same subject. Exclusion criteria included pregnancy or breastfeeding, serious systemic diseases (e.g., diabetes, thyroid problems, uncontrolled arterial hypertension, cardiovascular or pulmonary diseases), serious psychiatric conditions (e.g., depression, schizophrenia, bipolar disorder), skin pathologies at the site of stimulation (in particular, hand thenar eminence), inability to reliably rate pain, current use of drugs (e.g., analgesics, antihistamines, sodium or calcium channel blockers, antidepressants, narcotics or any tobacco products) that could interfere with pain sensations, bad sleep quality the night before the test, contraindications to rTMS (i.e., previous severe head trauma or neurosurgical intervention, epilepsy, active brain tumour, and implanted ferromagnetic devices, e.g., cardiac pacemaker, neurostimulator or cochlear implants). As a screening for polyneuropathy, all subjects underwent a clinical assessment of motor and sensory functions (including evaluation of tactile, pinprick, and vibratory sensations on hands and feet) that was normal. Participants were asked to avoid alcohol consumption and intense exercise for 24 h before the test.

Because of the exploratory nature of the study, the sample size was not formally computed but was decided a priori based on resources available and the literature’s evidence. In particular, Taylor et al. [24] showed that single rTMS sessions applied to the human cortex can exert analgesic effects in a population of 24 healthy subjects.

### 2.2. Study Procedures

All subjects underwent three different experimental sessions on separate days, at least one week apart from each other. In all sessions we first assessed the HPT and subsequently applied three constant trials (CT) and three OA trials at the thenar eminence of the right hand. The three sessions were applied in a balanced randomized order by using a computer-based algorithm, and included a baseline session, in which only quantitative sensory testing (QST) paradigms (i.e., HPT, CT and OA) were performed, and two rTMS sessions, in which the same QST measures were evaluated immediately after sham (placebo) or active rTMS.

#### 2.2.1. Quantitative Sensory Testing Assessment

QST examinations were performed in a quiet room with temperature maintained at 22–24 °C by two well-trained neurophysiology technicians. Before the investigation, participants familiarized themselves with the visual analogue scale (VAS) scoring system and the experimental procedures by having the tests demonstrated on their left hand. The experimental tests were performed using a 30 × 30 mm air-cooled heat probe capable of delivering a constant temperature in the range of 20 to 50 °C using a ramp (0.1–2 °C/s) and hold strategy. The probe was connected to a Q-sense Conditioned Pain Modulation device (Medoc, Ramat Yishai, Israel). Before starting all the experimental procedures, conditions of abnormally low or high local skin temperature values were ruled out by using an infrared thermometer. The skin temperature was in the range between 34 °C and 37 °C in the right hand thenar in each tested subject.

The HPT was assessed by means of the ‘method of limits’ following standardized procedures [25,26]. Increasing warm stimuli, directed from adaptation range towards heat pain sensation range, were applied. The patients were instructed to press a mouse button with the free left hand as soon as the temperature caused a peak VAS of 50–60 mm. The test was repeated three times to be certain of a consistent determination of the test temperature. Average values were calculated to obtain a single threshold score. Interstimulus intervals of 8–10 s were kept, and no clues were given to the subject at stimulus onset. In all trials, thermode adaptation temperature was set to 32 °C, with rates of temperature change of 1 °C/s.

After HPT determination, three OA trials and three constant trials were applied to the right hand thenar using a well-established procedure [15,27,28,29,30,31]. Average values were calculated to obtain single scores for OA and constant trials. OA trials included a 5 s interval of the individualized average HPT (T1), 5 s at a temperature 1 °C higher than T1 (T2), and 20 s at the same temperature as T1 (T3). Constant trials included 30 s of a stimulus at average HPT. For subjects with average HPT greater than 48 °C, stimulation intensity was set at 48 °C for both CT and OA (T1 and T3). Interstimulus intervals between trials were at least 30 s. Increase and decrease rates were 2 °C/s and 1 °C/s, respectively. The six trials were performed according to two pseudorandomized sequences (i.e., OA-CT-OA-OA-CT-CT and CT-OA-CT-CT-OA-OA). For all trials, participants were asked to evaluate the intensity of pain by using a continuous analogue-to-digital converter of VAS (CoVAS, Medoc, Israel) anchoring at 0 = ‘no pain’ and 100 = ‘the most intense pain imaginable’. Participants were naïve regarding the details of the OA paradigm or the study aims. All participants were instructed to carefully evaluate even small differences in pain.

Throughout the experiment, the thermode was firmly fixed on the thenar eminence by using an elastic band with velcro. Care was taken in placing the probe, such that the best contact between probe and skin surface was achieved without causing a feeling of constriction in the patient. Subjects were informed about how to respond correctly, and, during the test, they could not see the change in temperatures on the computer screen.

#### 2.2.2. Stimulation Procedures

Magnetic stimulation was performed using a figure-of-eight coil with a 70 mm diameter connected to a Magstim Rapid2 Stimulator (Magstim Company Ltd., Whitland, UK). All subjects were comfortably seated on a reclining chair and told to be as relaxed as possible during the entire rTMS session. They wore earplugs and a tight-fitting plastic swimming cap to mark the optimum stimulation site and ensure optimum coil placement. For active stimulation, the coil was placed with posteroanterior orientation over the optimal site for eliciting responses in the contralateral abductor pollicis brevis (APB) muscle [32]. For the sham stimulation, the coil was tilted 90° off the scalp with one wing touching the scalp over the same site as active rTMS. This has been described as a valid sham method to simulate ancillary aspects of TMS (acoustic artifact, contact with the scalp) without inducing biological effects [33]. Throughout the experimental session the subjects could not observe the position of the coil. The rTMS session consisted of 30 trains of TMS pulses delivered at 10 Hz for 4 s (40 pulses/train) with a 26 s intertrain interval, resulting in 1200 pulses per session for a total duration of 15 min. The stimulation intensity was set to 90% of the resting motor threshold for eliciting responses in the relaxed APB muscle. The resting motor threshold was defined as the minimum stimulation intensity needed to produce responses of at least 50 μV in at least 50% of ten trials. The subjects were given audiovisual feedback of electromyographic (EMG) activity to help maintain complete muscle relaxation. EMG signals were recorded using pre-gelled disposable surface Ag/AgCl electrodes (Ambu Neuroline 715 Surface Electrodes) placed 3 cm apart over the belly and tendon of the muscle. EMG activity was recorded with a bandpass of 10 to 1000 Hz and a display gain ranging from 50 to 200 μV/cm by using a Cadwell Sierra Summit EMG System (Cadwell Industries, Inc., Kennewick, WA 99336 USA). The coil position was continuously monitored throughout the experiment to keep it constant. Stimulation was performed following safety guidelines [34].

### 2.3. Statistical Analyses

Parametric statistics were used as data were normally distributed as assessed by the Kolmogorov–Smirnov test. A one-way analysis of variance (ANOVA) was carried out to assess differences among HPT values recorded in the three experimental sessions. Two-way repeated measures ANOVA with within-subjects factors ‘condition’ (two levels: OA and constant trial) and ‘time’ (30 levels: VAS score at every second during T1, T2 and T3) was chosen for its ability to explore differences among repeated measurements of VAS scores recorded during OA and constant trials at baseline and after active or sham rTMS. To disentangle adaptation or sensitization phenomena and offset effects, the magnitude of OA was calculated by subtracting pain ratings at 1 s time intervals assessed during the OA trial from the constant trial (ΔOA) [27]. After that, a two-way ANOVA with factors ‘session’ (three levels: baseline, active, and sham) and ‘time’ (20 levels: VAS assessment at every second during T3) was carried out to disclose any possible differences in ΔOA between sessions. Finally, a two-way ANOVA with factors ‘session’ (three levels: baseline, active, and sham) and ‘time’ (30 levels: VAS assessment at every second during T1, T2 and T3) was carried out to find possible between-session differences in VAS values recorded during the constant trials (i.e., differences in adaptation or sensitization phenomena). The Duncan’s post hoc test for multiple comparisons was used after ANOVA. The Duncan post hoc test was preferred to other more conservative post hoc tests considering the exploratory nature of the study. For all analyses, *p* < 0.05 was considered as significant.

Individual analyses were also carried out to evaluate the percentage of subjects showing a robust OA phenomenon at baseline and after active or sham stimulation. As a criterion for robust OA, we arbitrarily established a priori (before carrying out the statistical analyses) a decrease of at least 30 mm in the VAS score during the OA trial compared to the constant trial (i.e., ΔOA ≥ 30 mm) for at least 5 consecutive seconds during T3. The proportion of patients with or without robust OA phenomenon in the different experimental sessions was compared by means of the pairwise McNemar test with Bonferroni correction. The McNemar test was chosen because of the paired- sample experimental design. Statistical analyses were performed using the Statistical Package for the Social Sciences software (version 22, SPSS, Chicago, IL, USA).

## 3. Results

All enrolled subjects completed the planned experimental evaluations. Both rTMS and QST paradigms were well tolerated. No side effects occurred during or after the end of rTMS sessions.

No significant differences were found between different experimental sessions as regards the HPT values (Figure 1). Individualized mean HPT values ranged from 39 °C to 50 °C in baseline, from 42.2 °C to 50 °C after sham rTMS, and from 41.8 °C to 49.4 °C after real rTMS. Constant and OA (T1 and T3) trials were set at the individualized mean HPT value.

ANOVA performed to test differences in the VAS scores recorded during the OA and constant trials at baseline showed a significant interaction of factors (*p* < 0.0001). In the post hoc analysis, as expected, we recorded significant increased VAS values during the OA vs. constant trial (from the 10th to the 19th s, *p* < 0.05) due to the 1-degree increase in temperature at T2. Instead, no significantly reduced VAS values during the T3 were observed during the OA trial, indicating the absence of the OA phenomenon (Figure 2).

Similar findings were found during the sham session, as we observed significant interaction of factors (*p* < 0.0001) and, at the post hoc analysis, only significantly increased VAS values during the OA vs. constant trial (from the 8th to the 18th s, *p* < 0.05) (Figure 3).

During the active session, again we recorded a significant interaction of factors (*p* < 0.0001), and increased VAS values during the OA vs. constant trial (from the 8th to the 15th s, *p* < 0.05). But differently from the baseline and sham sessions, we also recorded significantly reduced VAS values during the OA vs. constant trial at T3, from the 25th to the 30th s (*p* < 0.05), indicating a delayed OA phenomenon (Figure 4).

Though higher ΔOA values were recorded at T3 during the active session (Figure 5), ANOVA did not show a significant interaction of factors.

No significant differences in VAS values recorded during the constant trial were found among sessions. The maximum value of VAS during the constant trial was recorded at the 27th s during baseline and sham sessions, and at the 30th s during the real session.

Individual analyses showed that a robust OA phenomenon could be seen only in four out of the 24 subjects (17%) after active rTMS, but in none at baseline or after sham rTMS. The McNemar test showed a significant difference among sessions (*p* < 0.001).

## 4. Discussion

To our knowledge, this is the first study assessing the effects of rTMS on the HPT and on the response to constant and dynamic heat pain stimuli in healthy subjects. The first main finding of the study was the lack of a significant OA phenomenon at the thenar eminence of healthy subjects, at least under baseline conditions. We showed, indeed, that a delayed and attenuated OA phenomenon can be observed in the same cohort after a single session of high-frequency rTMS. Thus, if on the one hand we confirmed the previous observation by Naugle et al. [19] that OA is normally not detectable at the glabrous skin of the hand, on the other hand we showed that the OA phenomenon in this body area could be observed under specific conditions. Considering that OA is thought to be primarily mediated by central mechanisms involving the descending pain modulatory and the reward systems, we hypothesize that the activation of these structures by rTMS may be responsible for our results [14,16]. Yelle et al. [35] found the activation of the periaqueductal grey (PAG) during OA by using functional MRI. The PAG is considered a pivotal centre of the brainstem for the top-down modulation of pain towards the dorsal horns of the spinal cord, where nociceptive information from the primary afferent neurons can be blocked. In another functional MRI study, the attenuation of OA was associated with inactivity of the areas of both the descending pain modulatory and reward systems in patients with chronic pain [16]. We cannot exclude that rTMS could have also interfered with the nociceptive signal transmission to the somatosensory cortex via the spinothalamic–thalamocortical pathway. However, this hypothesis is not supported by the finding that HPT did not significantly change after active rTMS in our study, differently from what was described by Johnson et al. [4] in patients with chronic pain.

Two lines of evidence should be considered to explain the absence of OA at the thenar eminence under baseline conditions together with the delayed and attenuated reappearance of OA after rTMS, differently from what was generally observed at the forearm [15,27,28,29,30,31]. First, the thickness of the epidermal stratum corneum of the glabrous skin of the palm is at least twice that in hairy skin, meaning that a thicker epidermal layer is interposed between nociceptors and skin surface [21,22]. Thus, when contact heat is applied by a thermode, it may take a longer time to reach the nociceptor activation threshold, or an optimal heat ramp profile could not be obtained to induce a robust OA phenomenon [20]. These suggestions could explain the absence or the attenuated and delayed OA responses. Indeed, when tested on the forearm, VAS reduction during the OA paradigm was clearly observed during the first 5–10 s of T3 [15,27,28,29,30,31], whilst at the hand thenar, we found a significant VAS reduction after active rTMS only at the end of the 20 s T3 time interval. The second aspect to be considered refers to possible differences in nociceptor innervation between the thenar eminence and the forearm [19,36]. In primates, the transmission of noxious mechanical and heat stimuli to the spinal cord is mediated by A-fibre mechano-heat (AMH) and C-fibre mechano-heat (CMH) skin nociceptors, which are responsible for first and second pain sensations, respectively [19,36,37]. Two populations of AMH nociceptor were described: type I (AMH-I) afferents, characterized by a high heat threshold (median 53 °C) with a slow, wind-up response to heat (in the order of seconds); and type II (AMH-II) afferents, characterized by a low heat threshold (median 46 °C) and a short latency (in the order of tens of milliseconds). Whereas AMH-I and AMH-II afferents are similarly represented in the hairy skin [37], experimental evidence suggests that AMH-II afferents may be absent in the glabrous skin of primates [36,38,39]. As the AMH-II nociceptors are thought to be primarily involved in the induction of inhibitory temporal sharpening mechanisms underlying OA, their absence could explain why OA is not easily detectable at the thenar eminence [19,35]. It is noteworthy, however, that this view has been criticized. Iannetti et al. [20] showed that laser stimulation of hairy and glabrous skin elicits very similar psychophysical ratings and electrophysiological responses. These authors therefore provided strong evidence that first pain to heat also exists in glabrous skin, and it is mediated by nociceptive afferents having the same physiological properties of AMH-II. Finally, it should be considered that the thenar eminence is highly innervated by C-fibre nociceptors [40] that are involved in the transmission of late pain and burning sensations [41,42]. Previous studies have also shown that the area of the receptive fields of C-fibres is much smaller on the hand with respect to other body sites [43,44]. This could explain why, in the present study, the VAS values progressively increased during the constant trial, reaching the maximum peak during the last seconds of stimulation. Indeed, differently from what was observed at the forearm, where adaptation phenomena prevail [27,28,29,30,31], sensitization phenomena mainly mediated by the CMH afferents could be predominant at the thenar eminence [23]. In turn, it has been hypothesized that sensitization could mask or delay the onset of OA [19]. In this regard, it should be mentioned that increased latencies to maximal OA were observed in conditions of experimentally induced sensitization (i.e., capsaicin-heat or heat-only sensitization) on the forearm, though in the absence of changes in the magnitude of OA phenomenon [30]. Based on these considerations, it is possible to hypothesize that the mechanisms underlying OA are largely different from those responsible for sensitization, although they could interact with each other. From an evolutionary point of view, the differences between the hand and other body sites such as the forearm regarding the response to both CT and OA remain unclear, being possibly linked to functional specificities of the hand.

## 5. Limitation and Conclusions

Some considerations and limitations of the study deserve discussion. Firstly, since a sham coil was not used, it was not possible to establish a strict control condition. Nevertheless, the role of possible placebo effect induced by rTMS seems unlikely considering that all subjects were unaware of the specific study aims and were naïve for both TMS and QST procedures. Furthermore, any placebo effect could have more easily affected the HPT rather than the response to the more complex OA paradigm. Secondly, the hand motor hotspot was targeted without neuronavigation; thus, we cannot exclude that even more evident changes could be observed after active rTMS by targeting the anterior border of the central sulcus using image-guided navigation. However, there is no current evidence that neuronavigation has any impact on the outcome of pain therapy when rTMS is applied to M1 [17]. Thirdly, reliability of the OA paradigm was not tested in this study, though evidence has been provided by other authors that this may be good to excellent [45,46]. Finally, future studies should confirm the modulatory effects of rTMS on OA in the hairy skin, e.g., in the forearm, where OA is normally detected. In a such condition, we should also consider that rTMS could fail in enhancing OA phenomenon, as it could already be optimally expressed in baseline conditions in healthy subjects. Therefore, studies in patients with chronic pain, in which the OA phenomenon has been shown to be compromised [27,28], could provide important information. Based on the above consideration, it is noteworthy to remark that the present results in healthy subjects, in which the mechanisms and anatomical circuits subserving OA phenomenon are normally functioning, cannot be directly translated to patients with different chronic pain syndromes. Future studies are needed to be carried out in the future to understand whether the M1 can be a good target for repeated rTMS sessions for treatment of different chronic pain conditions, such as migraine or fibromyalgia. Studies are also needed to evaluate whether alternative targets such as the dorsolateral prefrontal cortex might lead to different results or whether demographic factors such as gender and age may influence response to rTMS.

In conclusion, here we provided evidence that rTMS might activate central mechanisms underlying OA phenomenon in healthy subjects. Our results shed light on the mechanisms by which rTMS applied to M1could exert its antinociceptive action, fostering research on the therapeutic use of rTMS in patients with chronic pain.

## Figures and Tables

**Figure 1 jcm-12-07066-f001:**
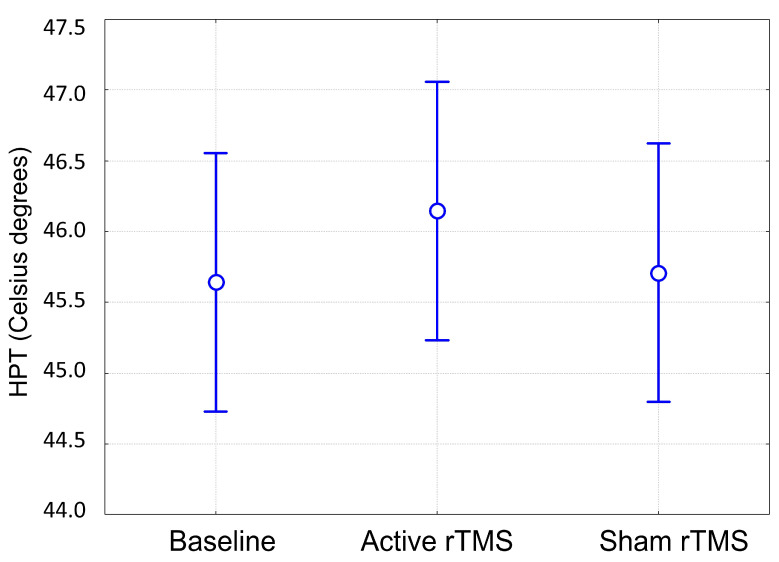
HPT values recorded at baseline and after active or sham rTMS. Vertical bars denote 95% confidence intervals. HPT: Heat Pain Threshold. rTMS: repetitive Transcranial Magnetic Stimulation.

**Figure 2 jcm-12-07066-f002:**
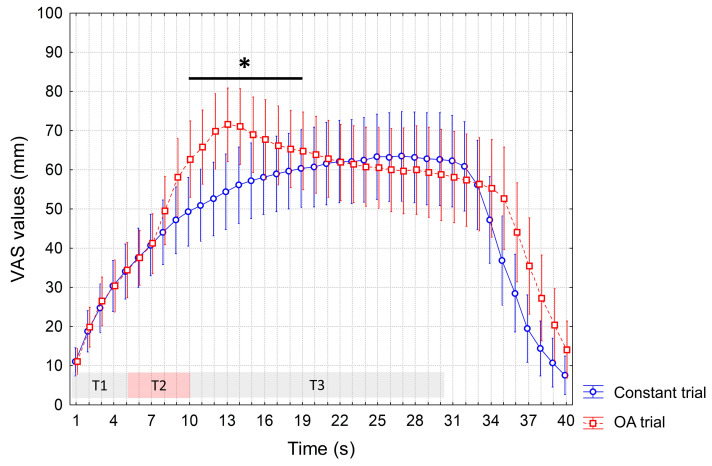
VAS values recorded during the constant trial and OA trial at baseline session. Mean values are reported. Vertical bars denote 95% confidence intervals. T1, T2 and T3 indicate different time intervals of OA trial, i.e., T1 = first 5 s interval of the individualized HPT, T2 = 5 s interval at a temperature 1 °C higher than T1, T3 = 20 s interval at the same temperature as T1. Constant trial included 30 s of a stimulus at HPT. The horizontal bar with an asterisk indicates significant differences (*p* < 0.05). HPT: Heat Pain Threshold. OA: offset analgesia. VAS: visual analogue scale.

**Figure 3 jcm-12-07066-f003:**
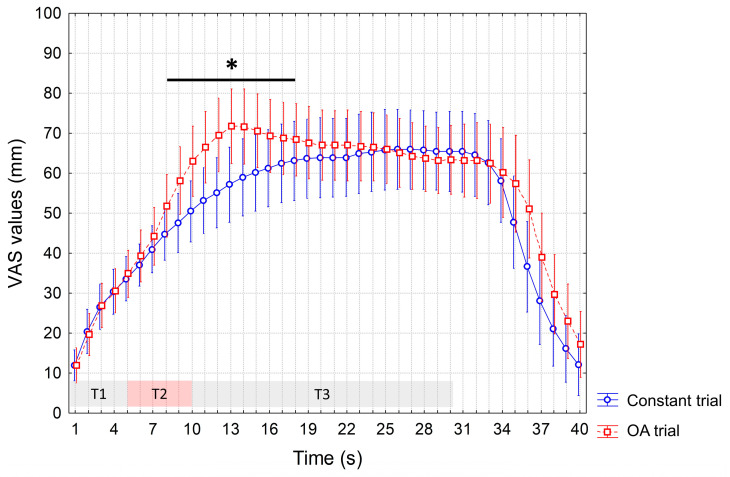
VAS values recorded during the constant trial and OA trial after sham rTMS. Mean values are reported. Vertical bars denote 95% confidence intervals. T1, T2 and T3 indicate different time intervals of the OA trial, i.e., T1 = first 5 s interval of the individualized HPT, T2 = 5 s interval at a temperature 1 °C higher than T1, T3 = 20 s interval at the same temperature as T1. Constant trial included 30 s of a stimulus at HPT. The horizontal bar with an asterisk indicates significant differences (*p* < 0.05). HPT: Heat Pain Threshold. OA: offset analgesia. rTMS: repetitive Transcranial Magnetic Stimulation. VAS: visual analogue scale.

**Figure 4 jcm-12-07066-f004:**
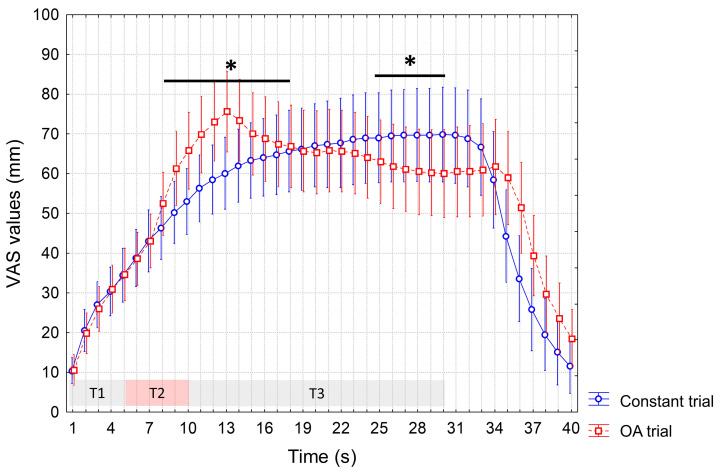
VAS values recorded during the constant trial and OA trial after active rTMS. Mean values are reported. Vertical bars denote 95% confidence intervals. T1, T2 and T3 indicate different time intervals of the OA trial, i.e., T1 = first 5 s interval of the individualized HPT, T2 = 5 s interval at a temperature 1 °C higher than T1, T3 = 20 s interval at the same temperature as T1. Constant trial included 30 s of a stimulus at HPT. The horizontal bars with an asterisk indicate significant differences (*p* < 0.05). HPT: Heat Pain Threshold. OA: offset analgesia. rTMS: repetitive Transcranial Magnetic Stimulation. VAS: visual analogue scale.

**Figure 5 jcm-12-07066-f005:**
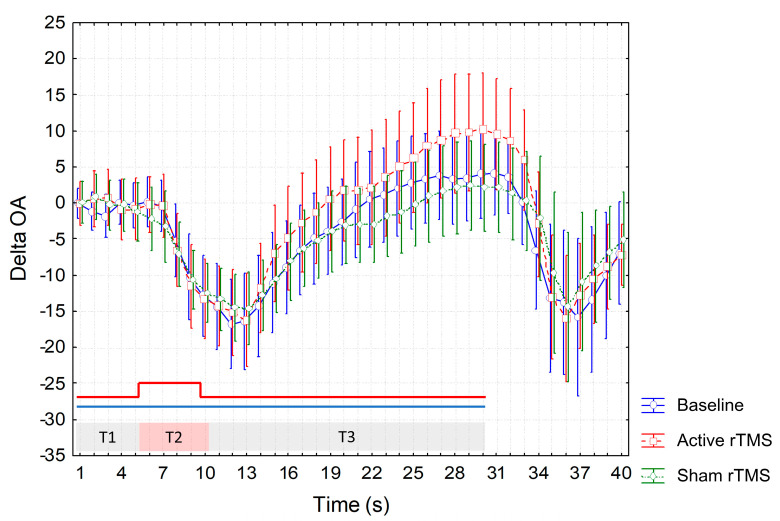
Magnitude of OA calculated by subtracting VAS values at 1−s time intervals assessed during OA trial from the constant trial (ΔOA) at different experimental sessions. Mean values are reported. Vertical bars denote 95% confidence intervals. T1, T2 and T3 indicate different time intervals of the OA trial, i.e., T1 = first 5 s interval of the individualized HPT, T2 = 5 s interval at a temperature 1 °C higher than T1, T3 = 20 s interval at the same temperature as T1 (red line at the bottom of the graph indicates temperature changes during the OA trial). Constant trial included 30 s of a stimulus at HPT (blue line at the bottom of the graph showing no temperature variations during the trial). Values below and above 0 indicate higher and lower VAS values during OA trial compared to the constant trial, respectively. HPT: Heat Pain Threshold. OA: offset analgesia. VAS: visual analogue scale.

## Data Availability

Raw data used in this study are available in the Zenodo repository: https://doi.org/10.5281/zenodo.8337701 (accessed on 12 September 2023).

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
