# Peer review of "Repetitive Transcranial Magnetic Stimulation of the Human Motor Cortex Modulates Processing of Heat Pain Sensation as Assessed by the Offset Analgesia Paradigm"

_jcm, 2023, doi:10.3390/jcm12227066_

Round 1
Reviewer 1 Report
Comments and Suggestions for Authors
I appreciate authors for this study, however, I have several concerns as follow:
· Rationale for Studying Healthy Subjects: The study investigates the effects of rTMS on healthy subjects. It would be important to discuss the relevance of studying healthy individuals in the context of understanding the mechanisms of pain relief for patients with chronic pain conditions. Are there any limitations associated with extrapolating findings from healthy subjects to patients with chronic pain?
· Sample Size and Demographics: The study includes 24 right-handed healthy volunteers. It's essential to discuss whether this sample size is adequate for drawing meaningful conclusions. Additionally, it would be valuable to provide more information about the demographic characteristics of the participants, such as age, gender distribution, and any potential biases introduced by these factors.
· Exclusion Criteria: The study lists several exclusion criteria for participant selection. Could the authors elaborate on the rationale behind each criterion and whether any potential biases or confounding variables were considered?
· Blinding and Sham Stimulation: The study mentions the use of sham stimulation. Could the authors provide more details on how blinding was maintained during the sham stimulation sessions to ensure that participants were unaware of the treatment condition?
· Quantitative Sensory Testing (QST) Parameters: It's important to provide specific details about the QST procedures, such as the exact temperature ranges used for assessing HPT, the criteria for determining OA, and the validity and reliability of these measurements.
· Statistical Analyses: The statistical methods used in the study are described briefly. Could the authors provide more information about the statistical tests used, including assumptions made, the rationale for selecting these tests, and the significance level?
· OA Phenomenon: The study aims to assess the OA phenomenon in the right thenar eminence. Given that the mechanisms of OA are not completely understood, it would be helpful to discuss the potential physiological basis of this phenomenon and how it relates to the study's objectives.
· Clinical Implications: The study focuses on understanding the mechanisms of pain relief induced by rTMS. Could the authors discuss the potential clinical implications of their findings for the treatment of chronic pain conditions and how this research may contribute to improving patient outcomes?
· Ethical Considerations: It's essential to ensure that ethical considerations, such as informed consent, were appropriately addressed in the study. Were participants fully informed about the nature of the study, including potential risks and benefits?
· Generalizability: Discuss the generalizability of the study's findings. Are there any limitations to extrapolating the results to a broader population, including patients with different chronic pain conditions or other demographics?
· Baseline OA Absence: The absence of a significant OA phenomenon at the thenar eminence under baseline conditions is an interesting finding. Could the authors elaborate on why OA is not easily detectable at this site compared to other areas of the body? Are there any hypotheses about the underlying mechanisms for this difference?
· Nociceptor Innervation: The study briefly mentions differences in nociceptor innervation between the thenar eminence and other skin areas. Could the authors expand on how these differences may affect the response to OA and how rTMS might interact with these nociceptor populations?
· Sensitization: The study suggests that sensitization phenomena in the thenar eminence could mask or delay the onset of OA. Could the authors discuss the potential implications of sensitization in the context of pain management and how rTMS might interact with sensitization processes?
· This study needs a separate section for Conclusion.
· Effect size for all statistics
· The link for row data is not working.
Comments on the Quality of English Language
The language is acceptable, but there is room for improvement.
Author Response
Reviewer 1
I appreciate authors for this study, however, I have several concerns as follow:
- Rationale for Studying Healthy Subjects: The study investigates the effects of rTMS on healthy subjects. It would be important to discuss the relevance of studying healthy individuals in the context of understanding the mechanisms of pain relief for patients with chronic pain conditions. Are there any limitations associated with extrapolating findings from healthy subjects to patients with chronic pain?
Answer: First of all, we are very grateful to the reviewer for his careful work and suggestions that allow us to significantly improve the work, highlighting limitations and providing clues for future research.
Regarding the first question, we agree with the referee comment. Our results in the healthy subjects cannot be directly translated to patients with different chronic pain syndromes, though the evidence that the mechanisms underlying OA can be enhanced in HS might provide a rationale for studies conducted in pathological conditions. It would also be important to study different pathological chronic pain conditions in which the mechanisms underlying OA could be dysfunctional (e.g., inhibited or insufficiently activated) or altered due to anatomical damage to the underlying neural network.
The following sentence has been added within the text in the Discussion: “However, it is noting that the present results in healthy subjects, in which the mechanisms and anatomical circuits subserving OA phenomenon are normally functioning, cannot be directly translated to patients with different chronic pain syndromes”.
- Sample Size and Demographics: The study includes 24 right-handed healthy volunteers. It's essential to discuss whether this sample size is adequate for drawing meaningful conclusions. Additionally, it would be valuable to provide more information about the demographic characteristics of the participants, such as age, gender distribution, and any potential biases introduced by these factors.
Answer: Because of the exploratory nature of the study, sample size was not formally computed but was decided a priori based on resources available and literature evidence. In particular, Taylor et al. (2012) showed that single rTMS sessions applied to the human cortex can exert analgesic effects in a population of healthy subjects of the same size as ours (i.e., 24 subjects). Regarding chronic pain conditions, single rTMS sessions have been proven to significantly modulate pain even in smaller samples of subjects (Lefaucheur et al., 2001, “Interventional neurophysiology for pain control: duration of pain relief following repetitive transcranial magnetic stimulation of the motor cortex”).
The above was specified within the text (Material and methods – subjects) and the following bibliographical reference has been added:
Taylor JJ, Borckardt JJ, George MS. Endogenous opioids mediate left dorsolateral prefrontal cortex rTMS-induced analgesia. Pain. 2012 Jun;153(6):1219-1225.
The demographic characteristics of the enrolled subjects are specified in the “Subjects” paragraph (Material and methods) as follows: “13 females; mean age ± standard deviation: 26 ± 6.9 years; age range: 20–55 years”. We agree that demographic factors could play a relevant role in this type of study. For this reason, to make the sample as homogeneous as possible, elderly patients in whom OA has been demonstrated to be reduced were not included in this study.
- Exclusion Criteria: The study lists several exclusion criteria for participant selection. Could the authors elaborate on the rationale behind each criterion and whether any potential biases or confounding variables were considered?
Answer: In accordance with previous studies assessing pain modulatory mechanisms, we have tried to exclude all relevant factors capable of modulating pain perception even if we certainly cannot exclude some potential biases and confounding factors. For example, although psychiatric pathologies were excluded based on the medical history, individual psychological factors may have played a role. Changes in cortical excitability linked to the menstrual cycle could represent another confounding factor in female subjects. In this study, women were not examined during their menstrual cycle, and we paid attention to carrying out the 3 experimental sessions in the same period of the menstrual cycle in the same subject. This latter information was added within the text in the “Subjects” paragraph (Material and methods).
- Blinding and Sham Stimulation: The study mentions the use of sham stimulation. Could the authors provide more details on how blinding was maintained during the sham stimulation sessions to ensure that participants were unaware of the treatment condition?
Answer: Since we don't have a sham coil in our lab, for the sham stimulation the coil was tilted 90° off the scalp with one wing touching the scalp over the same site as active rTMS. This has been described as a valid sham method to simulate ancillary aspects of TMS (acoustic artifact, contact with the scalp) without inducing biological effects (Lisanby et al., 2001). Throughout the experimental session the subjects could not observe the position of the coil. The above information was added within the text (paragraph: Stimulation procedures).
However, we agree that this method of sham stimulation may result less intense subjectively (i.e., less scalp muscle stimulation), representing a potential problem considering the within-subject crossover designs of the study. This limitation has been recognised in the discussion as follows: “since a sham coil was not used, it was not possible to establish a strict control condition. Nevertheless, the role of possible placebo effect induced by rTMS seems unlikely considering that all subjects were unaware of the study aims and were naïve for both TMS and QST procedures. Furthermore, any placebo effect could have more easily affected the HPT rather than the response to the more complex OA paradigm”.
- Quantitative Sensory Testing (QST) Parameters: It's important to provide specific details about the QST procedures, such as the exact temperature ranges used for assessing HPT, the criteria for determining OA, and the validity and reliability of these measurements.
Answer: Details about the QST procedures (HPT, OA, CT) are reported in the paragraph “Quantitative sensory testing assessment”. In all trials, thermode adaptation temperature was set to 32 °C. Individualized mean HPT ranged from 39 °C to 50 °C in baseline, from 42.2 °C to 50 °C after sham rTMS, and from 41.8 °C to 49.4 °C after real rTMS. Constant trials and T1 and T3 of OA trials were set at the individualized mean HPT value. For subjects with mean HPT greater than 48 °C, stimulation intensity was set at 48 °C. This information was specified within the text. According to previous studies (e.g., Szikszay et al., 2020 “ Offset analgesia: somatotopic endogenous pain modulation in migraine”) OA was calculated by subtracting pain ratings at 1-s time intervals assessed during the OA trial from the constant trial (ΔOA). This was made to disentangle adaptation or sensitization phenomena and offset effects. As a criterion for robust OA, considering the magnitude of the average OA observed in previous studies, we arbitrarily established a priori (before carrying out the statistical analyses) a decrease of at least 30 mm in the VAS score during the OA trial compared to the constant trial (i.e., ΔOA ≥ 30 mm) for at least 5 consecutive seconds during T3.
It was not the objective of the study to evaluate reliability of the QST procedures, as now reported as a limitation of the study. However, reliability of offset analgesia has been investigated in previous studies with a variety of protocols and reported in many cases to be good to excellent, though variable between studies (e.g., see de Vita et al., 2022; Nilsson et al., 2014). This information and new references (see below) were added withing the text.
New references:
- De Vita MJ, Buckheit K, Gilmour CE, Moskal D, Maisto SA. Development of a Novel Brief Quantitative Sensory Testing Protocol That Integrates Static and Dynamic Pain Assessments: Test-Retest Performance in Healthy Adults. Pain Med. 2022 Feb 1;23(2):347-351.
- Nilsson M, Piasco A, Nissen TD, Graversen C, Gazerani P, Lucas MF, Dahan A, Drewes AM, Brock C. Reproducibility of psychophysics and electroencephalography during offset analgesia. Eur J Pain. 2014 Jul;18(6):824-34.
- Statistical Analyses: The statistical methods used in the study are described briefly. Could the authors provide more information about the statistical tests used, including assumptions made, the rationale for selecting these tests, and the significance level?
Answer: The following information was added in the Statistical Analyses paragraph:
Repeated measures ANOVA with within-subjects factors ‘condition’ (two levels: OA and constant trial) and ‘time’ (30 levels: VAS score at every second during T1, T2 and T3) was chosen for its ability to explore differences among repeated measurements of VAS scores recorded during OA and constant trials at baseline and after active or sham rTMS.
Effect sizes from two-way repeated measure ANOVAs were reported using partial eta squared.
The sphericity assumption was checked by using Mauchly’s test, and Huynh–Feldt’s correction was adopted, if necessary, for the degrees of freedom.
The Duncan post hoc test was preferred to other more conservative post hoc tests considering the exploratory nature of the study.
The McNemar test was chosen because of the paired- sample experimental design.
For all analyses, p < 0.05 was considered as significant.
- OA Phenomenon: The study aims to assess the OA phenomenon in the right thenar eminence. Given that the mechanisms of OA are not completely understood, it would be helpful to discuss the potential physiological basis of this phenomenon and how it relates to the study's objectives.
Answer: As stated in the Introduction we chose to target the thenar eminence for different reasons (e.g. motor threshold values can be easily assessed in the hand muscles, a reliable coil placement can be achieved also without neuronavigation, optimal analgesic effects can be achieved when rTMS is applied over the corresponding somatotopic representation (within M1) of the painful territory) with the dual aim of evaluating whether OA is not actually recordable at this level and whether a neuromodulation paradigm is able to modulate the responses to the tested psychophysical paradigms (HPT, OA and CT).
We agree that the mechanisms underlying OA phenomenon are not well understood, and we know even less about what happens on the glabrous skin of the hand. Also in light of literature data, we believe that the absence of OA at the hand level and the tendency towards sensitization (rather than adaptation) during CT could mainly relay on differences in nociceptor innervation between the thenar eminence and the forearm, though other (not mutually exclusive) explanations cannot be excluded (such as greater thickness of the epidermis). In particular, the hand in highly innervated by C-fibre nociceptors, that are involved in the transmission of late pain and burning sensations and can be responsible for sensitization during the application of prolonged painful stimuli. Previous studies have also shown that the area of the receptive fields of C-fibres is much smaller on the hand (e.g. 6 mm2) with respect to other body sites, such as the toes (e.g., 35 mm2) or the lower leg (e.g., 213 mm2) (Van Hees and Gybels, 1981; Schmidt et al., 1997).
From an evolutionary point of view, the differences between the hand and other body sites such as the forearm regarding the response to both CT and OA remain uncertain, being possibly linked to functional specificities of the hand (we are tempted to speculate that in particular conditions it might be more useful to tolerate pain in the hand - unless it becomes potentially harmful to the tissues - rather than to move away from it, but this remains highly speculative). The text in the discussion has been enriched based on the above considerations and the following references have been added:
Schmidt R, Schmelz M, Ringkamp M, Handwerker HO, Torebjörk HE. Innervation territories of mechanically activated C nociceptor units in human skin. J Neurophysiol. 1997;78(5):2641-8.
Van Hees J, Gybels J. C nociceptor activity in human nerve during painful and non painful skin stimulation. J Neurol Neurosurg Psychiatry. 1981;44(7):600-7.
- Clinical Implications: The study focuses on understanding the mechanisms of pain relief induced by rTMS. Could the authors discuss the potential clinical implications of their findings for the treatment of chronic pain conditions and how this research may contribute to improving patient outcomes?
As mentioned in response to the reviewer's first comment, our data are not directly translatable to patients with chronic pain. However, demonstrating that antinociceptive mechanisms can be activated in healthy individuals may mean that rTMS could potentially at least partially restore them in patients in whom these mechanisms are dysfunctional. However, this remains hypothetical, and it cannot be excluded that in different chronic pain conditions the effect of rTMS could be different, both greater and lesser than what is observed in healthy people, also in relation to the type of functional and/or anatomical alteration of the nociceptive pathways and endogenous pain control mechanisms. Therefore, the study lays theoretical bases for the application of high-frequency rTMS to the M1 in different populations of patients with chronic pain but does not allow any conclusions to be drawn in this sense. This was made explicit in the “Limitation and conclusions” section.
- Ethical Considerations: It's essential to ensure that ethical considerations, such as informed consent, were appropriately addressed in the study. Were participants fully informed about the nature of the study, including potential risks and benefits?
As stated in the paragraphs “Institutional Review Board Statement” and “Informed Consent Statement” the study was conducted in accordance with the Declaration of Helsinki and approved by the local Ethics Committee. Patients were informed about the study procedures, the potential side effects of the methods used and the potential positive repercussions in patients with different pathological conditions with chronic pain. However, detailed information on the OA and CT trials was not provided to avoid the influence of voluntary or suggestion-related components on the results. This has been explained within the text.
- Generalizability: Discuss the generalizability of the study's findings. Are there any limitations to extrapolating the results to a broader population, including patients with different chronic pain conditions or other demographics?
Answer: as mentioned above, the discussion and conclusions were enriched as follows:
“[…] Therefore, studies in patients with chronic pain, in which the OA phenomenon has been shown to be compromised, could give important information. Based on the above consideration, it is to remark that the present results in healthy subjects, in which the mechanisms and anatomical circuits subserving OA phenomenon are normally functioning, cannot be directly translated to patients with different chronic pain syn-dromes. Future studies are needed to be carried out in the future to understand wheth-er the M1 can be a good target for repeated rTMS sessions for treatment of different chronic pain conditions, such as migraine or fibromyalgia. Studies are also needed to evaluate whether alternative targets such as the dorsolateral prefrontal cortex might lead to different results or whether demographic factors such as gender and age may influence response to rTMS”.
- Baseline OA Absence: The absence of a significant OA phenomenon at the thenar eminence under baseline conditions is an interesting finding. Could the authors elaborate on why OA is not easily detectable at this site compared to other areas of the body? Are there any hypotheses about the underlying mechanisms for this difference?
Answer: See response above (question 7).
- Nociceptor Innervation: The study briefly mentions differences in nociceptor innervation between the thenar eminence and other skin areas. Could the authors expand on how these differences may affect the response to OA and how rTMS might interact with these nociceptor populations?
- Sensitization: The study suggests that sensitization phenomena in the thenar eminence could mask or delay the onset of OA. Could the authors discuss the potential implications of sensitization in the context of pain management and how rTMS might interact with sensitization processes?
Answer: as mentioned above these aspects have been expanded also mentioning evidence from studies showing that the area of the receptive fields of C-fibres is much smaller on the hand with respect to other body sites. In this study we did not observe a modulation of the response to CT by rTMS, so we have no evidence to hypothesize a direct effect on the sensitization mechanisms that occur at the spinal level. As stated within the text it has been hypothesized that sensitization could mask or delay the onset of OA. In this regard, increased latencies to maximal OA were observed in conditions of experimentally induced sensitization (i.e., capsaicin-heat or heat-only sensitization) on the forearm, though in the absence of changes in the magnitude of OA phenomenon. Based on these considerations, it is possible to hypothesize that the mechanisms underlying OA are largely different from those responsible for sensitization, although they could interact with each other.
- This study needs a separate section for Conclusion.
Answer: A paragraph “Limitation and conclusions” has been now included.
- Effect size for all statistics.
Answer: Effect sizes from two-way repeated measure ANOVAs were reported using partial eta squared.
- The link for row data is not working.
Answer: the link will be active in case of acceptance of the work.
Comments on the Quality of English Language
The language is acceptable, but there is room for improvement.
Answer: the entire manuscript has been reviewed and checked for errors and typos.

Reviewer 2 Report
Comments and Suggestions for Authors
This study is a nicely presented investigation of the offset analgesia phenomenon in healthy subjects. The thenar eminence was selected as the site of testing and the main outcome metric was VAS during heat application. rTMS of M1 at 90% of motor threshold was the intervention. For sham rTMS “…the coil was tilted 90° off the scalp with one wing touching the scalp over the same site as active rTMS.”
The findings were that OA was not present in this healthy patient population and study design, however, with active rTMS, but not with sham, there appeared to be an emergence of OA. The authors conclude that rTMS at M1 might have antinociceptive effects and call for further study in the chronic pain population.
The authors reasonably address the significant limitations. In particular, they note that other sites of testing such as the forearm might yield different results and that the sham was not rigorous.
Nonetheless a provocative study.
Author Response
Reviewer 2
This study is a nicely presented investigation of the offset analgesia phenomenon in healthy subjects. The thenar eminence was selected as the site of testing and the main outcome metric was VAS during heat application. rTMS of M1 at 90% of motor threshold was the intervention. For sham rTMS “…the coil was tilted 90° off the scalp with one wing touching the scalp over the same site as active rTMS.”
The findings were that OA was not present in this healthy patient population and study design, however, with active rTMS, but not with sham, there appeared to be an emergence of OA. The authors conclude that rTMS at M1 might have antinociceptive effects and call for further study in the chronic pain population.
The authors reasonably address the significant limitations. In particular, they note that other sites of testing such as the forearm might yield different results and that the sham was not rigorous.
Nonetheless a provocative study.
Answer: We warmly thank the reviewer for evaluating and appreciating our work.
Round 2
Reviewer 1 Report
Comments and Suggestions for Authors
Many thanks, most comments were addressed!